# Cross-cultural adaptation and measurement properties of the Malay Shoulder Pain and Disability Index

**Caryn-Ann Ho[1], Jerri Chiu Yun Ling[2], Samihah Abdul Karim**[3,4]*

**1** Sports Medicine Department, University Malaya Medical Centre, Kuala Lumpur, Malaysia, **2** Sports Medicine Unit, Department of Orthopaedics, Hospital Tengku Ampuan Rahimah Klang, Selangor, Malaysia, **3** Sports Medicine Unit, Faculty of Medicine, University Malaya, Kuala Lumpur, Malaysia, **4** Biomedical Engineering Department, Faculty of Engineering, University Malaya, Kuala Lumpur, Malaysia

* samihahk@ummc.edu.my

## Abstract

### Objective

The purpose of this study is to cross-culturally adapt the Shoulder Pain and Disability Index from English to Malay, and to evaluate the measurement properties of the Malay version among Malay speakers with shoulder pain.

### Methods

Cross-cultural adaptation of the Malay version of Shoulder Pain and Disability Index (M-SPADI) was conducted according to international guidelines. 260 participants (Shoulder pain = 130, No shoulder pain = 130) completed the M-SPADI, the Numerical Rating Scale (NRS), and measurement of shoulder active range of motion (AROM). 54 participants repeated M-SPADI within a mean of 9.2 days.

### Results

Cross-cultural adaptation of M-SPADI had no major issues. The M-SPADI had good face validity; item and scale content validity indexes (I-CVI, S-CVI) were >0.79 except for Disability Item 3 (I-CVI = 0.75), and exploratory factor analysis showed that M-SPADI had a bidimensional structure. There was a strong positive correlation between M-SPADI and NRS ($r_{Pain}$ = 0.845, $r_{Disability}$ = 0.722, $r_{Total}$ = 0.795, p <0.001) and a negative correlation between M-SPADI and shoulder AROM with the following correlation ranges ($r_{Pain}$ = -0.316 to -0.637, $r_{Disability}$ = -0.419 to -0.708, $r_{Total}$ = -0404 to -0.697, p<0.001). M-SPADI's total score was higher in participants with shoulder pain (Mdn: 33.8, IQR = 37.3) compared to no shoulder pain (Mdn:0, IQR = 0.8) and the difference was statistically significant (U = 238.5, z = -13.89, p<0.001). M-SPADI had no floor or ceiling effects (floor/ceiling <15%), high internal consistency (Cronbach's $\alpha_{Pain}$ = 0.914, Cronbach's $\alpha_{Disability}$ = 0.945) and good to excellent test-retest reliability ($ICC_{Pain}$ = 0.922, $ICC_{Disability}$ = 0.859, $ICC_{Total}$ = 0.895).

**Data Availability Statement:** Data cannot be shared publicly because of it contains potentially identifying information of human subjects. Data are available from the RedCap UMMC Institutional Data Access / Ethics Committee (contact via email at

nahar@ummc.edu.my) for researchers who meet the criteria for access to confidential data.

**Funding:** -Samihah -PV2018-62 -UMSC Care Fund, Faculty of Medicine University of Malaya - https://umresearch.um.edu.my/funding-opportunities-forms - NO. The funders had no role in study design, data collection and analysis, decision to publish, or preparation of the manuscript.

**Competing interests:** The authors have declared that no competing interests exist.

## Conclusion

M-SPADI has a bi-dimensional structure with no floor or ceiling effects, established face, content and construct validity, internal consistency, and test-retest reliability. M-SPADI is a reliable and valid tool for assessing Malay-speaking individuals with shoulder pain in clinical and research settings.

## Introduction

Shoulder pain is a common musculoskeletal disorder with a lifetime prevalence of 7 to 67% [1, 2]. Associated symptoms include restricted shoulder motion, disturbed sleep, and impaired activities of daily living [3–5]. Globally, it causes work absence, disability, and increased healthcare costs [2, 6]. In Malaysia, shoulder injury is ranked third in musculoskeletal disorders causing disability and fourth in the total cost of workers' compensation claims per body part [7].

Health-related patient-reported outcome measures (PROM) are essential to patient-centred care and research [8]. There are at least 50 PROMs measuring shoulder function, of which the most frequently used in research include the Shoulder Pain and Disability Index (SPADI), the Constant-Murley Shoulder Scale, and the American Shoulder and Elbow Surgeons Society Standardized Shoulder Assessment Form [9]. Systematic reviews concluded that no single shoulder assessment tool was superior to the other but recommended SPADI as a tool for clinical and research use [9–11].

SPADI is an English, self-administered, shoulder-specific PROM developed by Roach et al. to measure pain and disability in patients with shoulder joint disorder [1, 9]. SPADI is short, easily understood, applicable in various shoulder pathology, the most responsive, and has established validity and reliability [9–11]. SPADI has been cross-culturally adapted into multiple languages, including Spanish, Chinese, Thai, Marathi, Brazilian-Portuguese, Greek, Italian, Telugu, Tamil, German, Turkish, Danish, Persian, Slovene, and Dutch languages [3–5, 12–23].

There is currently no shoulder-specific PROM that has been cross-culturally adapted to the Malay language. Studies have shown that individuals with limited English proficiency find English PROMs difficult to comprehend and challenging to complete [8]. In Malaysia, where fresh graduates and students have low English proficiency [24], a Malay version of SPADI (M-SPADI) would be most beneficial as Malay is the national language and is widely spoken. Moreover, it is a compulsory language taught in all primary and secondary schools [25].

Given SPADI's high clinical value and increased use in multicultural research, M-SPADI is needed for use in the Malay-speaking population. Our objective was to cross-culturally adapt the Shoulder Pain and Disability Index from English to Malay, and to evaluate the measurement properties of the Malay version among Malay speakers with shoulder pain. The measurement properties evaluated include face validity, content validity, structural validity, hypothesis testing for construct validity, known-group validity, floor and ceiling effects, internal consistency, and test-retest reliability. We reported this study following the COSMIN 2021 reporting guidelines [26].

## Materials and methods

### Study design

This study was a two-phase cross-cultural adaptation and validation of M-SPADI following standard guidelines [27–29]. We received permission to cross-culturally adapt the Original

SPADI to M-SPADI from Professor Roach and obtained study approval from the Medical Ethics Research Committee, University of Malaya Medical Centre (UMMC) (MREC ID: 2020513–8617). We conducted the study per the Declaration of Helsinki, and all participants gave written informed consent.

## Participants

Research participants for the shoulder pain group and the no shoulder pain group were recruited via universal sampling from patients attending the Sports Medicine Clinic, UMMC from 1st June 2020 to 12th May 2021. The inclusion criteria for the shoulder pain group were patients attending Sports Medicine Clinic with shoulder pain, $\geq$ 18 years old, and understood the Malay language. The inclusion criteria for the no shoulder pain group were patients attending Sports Medicine Clinic without shoulder pain, $\geq$ 18 years old, and understood the Malay language. For both shoulder pain and no shoulder pain group, the exclusion criteria were neck, elbow, wrist, or hand injury, decline to give consent, and psychiatric illness.

We calculated a sample size of 50 participants for the pretesting of M-SPADI and 60 participants for test-retest [27, 30]. Based on a participant to item ratio of 10: 1, the validation study sample size was 130 participants [28, 30]. To assess known group validity, another 130 participants with no shoulder pain were recruited [28, 30].

## Instruments

**Shoulder Pain and Disability Index.**   This study used the numerical rating scale version of SPADI. SPADI has 13 items subdivided into two subscales which measure pain (five items) and disability (eight items) [1]. The initial visual analog scale (VAS) was replaced by an 11-point numerical scale where the individual scored their level of pain or difficulty from 0 to 10 with the anchors 'no pain/ no difficulty' and ' worst pain imaginable/ so difficult it requires help' [31]. Based on COSMIN guidelines and Coltman et al., the author considered SPADI a reflective construct [31, 32]. The original SPADI development study reported good internal consistency with Cronbach $\alpha$ total = 0.95, pain = 0.86 and disability = 0.95 [1]. They also reported a moderate test-retest reliability with intraclass correlation coefficient (ICC) = 0.66 and moderate to high negative correlation between shoulder active range of motion (AROM) and SPADI scores (Pearson correlation coefficient range, r = -0.545 to -0.804, p<0.001) [1]. A recent systematic review reported that SPADI had a bidimensional structure, good internal consistency with Cronbach $\alpha$ ranging from 0.85–0.96, good to excellent test-retest reliability with ICC ranging from 0.89–0.92 and no floor or ceiling effects [9].

Each subscale score is calculated using the formula:

$$(sum\ of\ subscale's\ item\ score \div subscale's\ maximal\ possible\ score) \times 100$$

[1, 9]. The maximal possible score excludes any unmarked item but requires at least 3/5 pain items and 6/8 disability items answered for SPADI to be scored [10]. The total SPADI score is the unweighted mean of pain and disability domain scores [9]. The scores range from 0 = the best to 100 = the worst with no cut-off point to indicate severity as it was designed to measure current status and change over time [1, 9].

**The Numerical Rating Scale (NRS).**   The NRS is an instrument for pain intensity assessment where individuals are asked to select a number from 0 to 10 that best describes their pain intensity [33]. The anchors are zero for no pain and ten for the worst pain ever possible [33].

## Phase 1: Cross-cultural adaptation

The cross-cultural adaptation process consisted of translating SPADI from English to Malay and culturally adapting M-SPADI to the Malaysian culture following Beaton et al. 2000 guidelines [27, 34].

**Stage 1: Initial translation.** Three independent bilingual Malay native-speaking translators forward translated SPADI from English to Malay, producing FT1, FT2, and FT3. The naive translator was a Malay Language teacher, whereas the informed translators were an Associate Professor of Sports Medicine and a Sports Medicine Physician [27].

**Stage 2: Synthesis of the translation.** Two researchers synthesized FT1, FT2, and FT3 to produce FT4. Resolution of issues was by consensus, and the principal researcher acted as observer and scribe.

**Stage 3: Back translation.** Three independent translators back-translated FT4 from Malay to English producing BT1, BT2, and BT3. All translators were bilingual, had no medical background, were unfamiliar with the concepts explored in SPADI, and had no access to the original SPADI. The translators were an English lecturer and two English teachers.

**Stage 4: Expert committee.** A bilingual expert committee reviewed all the abovementioned versions of the questionnaire and, through consensus, consolidated the prefinal version of the M-SPADI. The expert committee consisted of a senior lecturer cum certified translator from the University Malaya Faculty of Languages and Linguistics, a Professor of Orthopaedic Surgery, a Professor of Family Medicine, two Sports Medicine Physicians, a Rehabilitation Medicine Physician, a Senior Physiotherapist, an Occupational Therapist, and an Exercise Physiologist. They also completed the content validity forms for content validity assessment.

**Stage 5: Pretesting of M-SPADI.** Fifty participants completed the prefinal version of M-SPADI and answered the following questions; What did they understand about the questions asked (Both items and response)? [27] Were the questions relevant to shoulder pain and disability? [30] Was there any difficulty understanding the questions? and Any suggestions for improvement? These answers were subsequently used to assess Face Validity. These participants were retained for the forthcoming validation study.

**Stage 6: Submission of documentation to developers.** We submitted all documentation and a report of the adaptation process to the original author.

## Phase II: Validation study (measurement property assessment)

130 participants with shoulder pain and another 130 participants without shoulder pain filled in a written consent form and the Research Electronic Data Capture (REDCap) online survey. The survey consisted of the Demographics form (S1 Appendix), the M-SPADI Questionnaire (S2 Appendix), and the NRS. A researcher measured the AROM of the affected shoulder using standard shoulder goniometer technique [35].

The first 60 participants who were not undergoing invasive procedures or starting new treatments in the week preceding and following their study enrolment date were asked to repeat an identical M-SPADI seven days later. This was to ensure that the participant's condition was stable during testing. 56 participants out of 60 (93%) returned the second M-SPADI.

## Data management

Study data were collected and managed using the REDCap electronic data capture tools hosted by the University of Malaya. REDCap is a secure, web-based software platform designed to support data capture for research studies [36]. It is compliant with the Health Insurance Portability and Accountability act [37]. Strategies used to minimize missing data included collecting only essential information and designing a user-friendly online survey that can only proceed

to the next page upon completing all items. Any survey which did not fulfil the SPADI scoring criteria was excluded from the study.

## Statistical analysis

Statistical evaluation of measurement properties was carried out using the IBM SPSS Statistics for Windows, version 22 (IBM Corp., Armonk, NY, USA). An α-level of 0.05 was used for all statistical tests as the significance cut-off point, and normality for all data was assessed using Q-Q plots and the Kolmogorov-Smirnov test. Non-parametric tests were performed when data were not normally distributed. All statistical analysis result classification keys were summarised in their respective tables.

**Validity.**   Content validity was evaluated by a panel of eight expert committee members using Content validity forms (S3 Appendix) during stage 4: Expert committee. We subsequently calculated the content validity index for each item (I-CVI) and the overall scale (S-CVI) [38].

The M-SPADI's face validity was assessed during the pretesting of M-SPADI. 50 participants gave feedback regarding what they understood about the questions, were the questions relevant, any difficulty understanding the questions, and any suggestions for improvement.

Structural validity was assessed by performing exploratory factor analysis (EFA) to determine the factor structure of M-SPADI. The extraction method was principal component analysis with varimax rotation and Kaiser normalization. Factors with eigenvalues ≥1 were extracted, and factor loading ≥0.5 was considered significant [39].

As there is no gold standard shoulder-specific questionnaire in the Malay language, criterion validity could not be tested [28]. Hypotheses testing for construct validity was assessed using convergent validity and known-group validity [28, 29]. Spearman's correlation was used to calculate the correlation between M-SPADI and the NRS and between M-SPADI and shoulder AROM. We hypothesized that there would be a strong positive correlation between the M-SPADI and the NRS. We also hypothesized that M-SPADI would have a negative correlation to shoulder AROM.

Mann-Whitney U test was performed to assess known-group validity by comparing rank means of M-SPADI scores for the shoulder pain and no-shoulder pain groups. We hypothesized that M-SPADI scores for the shoulder pain group would be higher than the no shoulder pain group, and the difference would be statistically significant.

Floor and ceiling effects were assessed and considered present if ≥15% of participants achieved the lowest or highest possible subscale or total scores [40].

**Reliability.**   We evaluated internal consistency by calculating Cronbach α for each M-SPADI unidimensional scale [26].

Test-retest reliability was assessed using the ICC with a 95% confidence interval based on average measurement, absolute agreement, 2-way mixed-effects model [29, 41, 42]. The selected interval between repeated measures was seven days to prevent recall but ensure no clinical change had occurred [29].

## Results

### Description of sample

Fifty participants enrolled in the pretesting of M-SPADI. The phase II validation study had 260 participants, consisting of the shoulder pain group (n = 130) and no shoulder pain group (n = 130). The demographic characteristics of the shoulder pain group and no shoulder pain group differed significantly in age (U = 5261, z = -5.26, p = 0.000). However, there was no statistically significant difference between gender and level of education between the shoulder

**Table 1. Participant demographic characteristics and descriptive statistics of M-SPADI, NRS, and shoulder active range of motion.**

| | Pretesting of M-SPADI (n = 50) | Validation Study | |
| --- | --- | --- | --- |
| | | Shoulder pain group (n = 130) | No shoulder pain group (n = 130) |
| Age, years (Median, IQR) | 36.7(28.3) | 55.8(24.6)[1] | 35.4(23.3)[1] |
| Male | 30(56.6%) | 63(48.5%) | 65(50%) |
| Female | 23(43.4%) | 67 (51.5%) | 65(50%) |
| **Occupation** | | | |
| Employed | 40 (75.4%) | 75 (57.7%) | 101(77.7%) |
| Retired | 10 (18.9%) | 34 (26.2%) | 15(11.5%) |
| Homemaker | 2 (3.8%) | 15(11.5%) | 6(4.6%) |
| Other | 1 (1.9%) | 6(4.6%) | 8(6.2%) |
| **Level of education** | | | |
| No formal education | 1 (1.9%) | 1(0.7%) | 0 (0%) |
| Primary education | 1 (1.9%) | 3(2.3%) | 2(1.5%) |
| Secondary education | 11 (20.6%) | 40(30.8%) | 23(17.7%) |
| Tertiary education | 40 (75.5%) | 86(66.2%) | 105(80.8%) |
| **Affected Shoulder** | | | |
| Dominant | - | 97(74.6%) | - |
| Non-Dominant | - | 33(25.4%) | - |
| **Stage** | | | |
| Acute (<6 weeks) | - | 11 (8.5%) | - |
| Subacute (6–12 weeks) | - | 20 (5.4%) | - |
| Chronic (>12 weeks) | - | 99 (76.1%) | - |
| **M-SPADI (Median, IQR)** | | | |
| Pain | 18.0(44.0) | 46.0(40.5) | 0(0) |
| Disability | 6.3(30.0) | 26.3(37.8) | 0(0) |
| Total Score | 11.5(35.0) | 33.8(37.3) | 0(0.8) |
| **NRS (Median, IQR)** | 2.0(6.0) | 5.0(5.0) | 0(0) |
| **Shoulder Active Range of Motion (Median, IQR)** | | | |
| Forward Flexion | 178.0(22.5) | 155.5(50.0) | 180.0(0) |
| Abduction | 178.0(35.0) | 150.0(75.5) | 180.0(0) |
| Extension | 55.0(15.0) | 55.0(15.0) | 60.0(0) |
| Internal Rotation | 55.0(40.0) | 40.0(40.0) | 70.0(20.0) |
| External Rotation | 80.0(37.5) | 50.0(53.3) | 90.0(10.0) |

M-SPADI: Malay Shoulder Pain and Disability Index, NRS: Numerical Rating Scale, IQR: Interquartile range,

[1]$p < 0.05$, statistically significant difference.

pain and no shoulder pain group ($p > 0.05$). Participant demographics and descriptive statistics of M-SPADI, NRS and shoulder AROM are reported in Table 1. Q-Q Plots and Kolmogorov-Smirnov tests ($p < 0.05$), revealed that data for M-SPADI scores, NRS scores and shoulder AROM were non normally distributed for the pretesting of M-SPADI group, the shoulder pain group and the no shoulder pain group. The shoulder pain diagnoses were rotator cuff injuries with or without impingement (54.6%), frozen shoulder (14.6%), labral injuries (3.1%), acromioclavicular joint disease (3.8%), and others (23.9%). The median time to complete M-SPADI was 2 minutes (range 1–5 minutes), comparable to previous studies. Due to the online survey design, there was no missing data except for test-retest, where four participants did not complete the second M-SPADI.

## Cross-cultural adaptation

There were no significant problems encountered in the process of cross-cultural adaptation of SPADI into Malay. Minor issues encountered during translation were due to the selection of synonyms for example, carrying→*mengangkat* or *memikul*. Several items required consensus from the expert committee, such as Pain Item 5 pushing with the involved arm →*menolak dengan lengan yang terlibat*. The sentence was confusing as one does not push with *lengan*, which means forearm. The item was then rephrased to *menolak dengan tangan yang sakit*. The term jumper in Disability Item 3 was correctly translated to *baju sejuk* but was questioned on its suitability for the Malaysian population. The expert committee decided to keep the term *baju sejuk* in the pretesting of M-SPADI. Analysis of the pretesting of M-SPADI revealed no issues faced by participants with the term *baju sejuk*.

## Validity

**Content validity.** The S-CVI and I-CVI for all items were >0.79 except for Disability Item 3 "putting on an undershirt or jumper", which had an I-CVI = 0.75 (Table 2). Since Disability Item 3's I-CVI score was between 0.70–0.79 it required amendment but not elimination. Following the content validity scoring, this item was amended by the expert committee and trialled during the pretesting of M-SPADI. Based on the pretesting of M-SPADI results the expert committee agreed to keep the item.

**Face validity.** The feedback from the pretesting of M-SPADI was good, with minimal issues in understanding the M-SPADI items and response by participants. All participants agreed that the questionnaire was clear and easy to understand (100%), and 86% gave positive feedback when asked if the questions were relevant to shoulder pain and disability. Overall M-SPADI has good face validity.

**Table 2. Content validity of M-SPADI.**

| ITEM | I-CVI: Consistency | I-CVI: Representative | I-CVI: Relevance | I-CVI: Comprehensibility |
|---|---|---|---|---|
| Pain subscale item 1 | 1 | 1 | 1 | 1 |
| Pain subscale item 2 | 1 | 1 | 1 | 1 |
| Pain subscale item 3 | 1 | 1 | 1 | 1 |
| Pain subscale item 4 | 1 | 1 | 1 | 0.87 |
| Pain subscale item 5 | 1 | 1 | 1 | 0.87 |
| S-CVI (Pain) | 1 | 1 | 1 | 0.95 |
| Disability subscale item 1 | 1 | 1 | 1 | 1 |
| Disability subscale item 2 | 1 | 1 | 1 | 0.87 |
| Disability subscale item 3 | 1 | 1 | 1 | 0.75 |
| Disability subscale item 4 | 1 | 1 | 1 | 0.87 |
| Disability subscale item 5 | 1 | 1 | 1 | 1 |
| Disability subscale item 6 | 1 | 1 | 1 | 1 |
| Disability subscale item 7 | 1 | 1 | 1 | 1 |
| Disability subscale item 8 | 1 | 1 | 1 | 1 |
| S-CVI (Disability) | 1 | 1 | 1 | 0.83 |
| S-CVI (Total score) | 1 | 1 | 1 | 0.94 |

M-SPADI: Malay Shoulder Pain and Disability Index

I-CVI: item content validity index, S-CVI: scale content validity index

I-CVI>0.79 (appropriate), I-CVI = 0.70–0.79 (requires revision), and I-CVI<0.70 (eliminated) [38]

**Table 3. Exploratory factor analysis of M-SPADI.**

| | Factor | |
|---|---|---|
| | **1 (Disability)** | **2 (Pain)** |
| Pain subscale item 1 | | 0.833 |
| Pain subscale item 2 | | 0.834 |
| Pain subscale item 3 | | 0.756 |
| Pain subscale item5 | | 0.765 |
| Disability subscale item 7 | | 0.659 |
| Disability subscale item 1 | 0.790 | |
| Disability subscale item 2 | 0.688 | |
| Disability subscale item 4 | 0.855 | |
| Disability subscale item 5 | 0.857 | |
| Disability subscale item 8 | 0.819 | |
| Pain subscale item 4 | 0.596 | 0.635 |
| Disability subscale item 3 | 0.738 | 0.506 |
| Disability subscale item 6 | 0.656 | 0.609 |
| Eigenvalues | 8.686 | 1.175 |
| Percentage of Variance | 66.813% | 9.039% |

Extraction method: Principal component analysis

Rotation method: varimax with Kaiser normalization

Based on a sample size of 130, significant factor loading is $\geq 0.50$

Underlined factor loading indicates the highest loading factor in cases of cross-loading

**Structural validity.** EFA was performed using principal component analysis with varimax rotation. The Kaiser-Meyer Olkin = 0.930 and Bartlett's test of sphericity were significant ($c^2_{(78)}$ = 1613.77, $p <$0.001). Two factors were extracted with eigenvalues >1 where factor 1 explained 66.81% of variance and factor 2 explained 9.04% of variance. Items that loaded onto factor 1 were Disability Items 1, 2, 4, 5, and 8, while items that loaded onto factor 2 were Pain Items 1, 2, 3, 5, and Disability Item 7. Pain Item 4, Disability Item 3, and 6 showed cross-loading, as demonstrated in Table 3.

**Hypotheses testing for construct validity.** *1. Convergent validity.* Spearman's correlation revealed a statistically significant, strong, and positive correlation between the NRS and M-SPADI pain subscale (r = 0.845, p<0.001), M-SPADI disability subscale (r = 0.722, p<0.001), and M-SPADI total scores (r = 0.795, p<0.001). The results supported our hypothesis that M-SPADI subscales and total scores are strongly and positively correlated with the NRS.

M-SPADI pain subscale (correlation range r = -0.316 to -0.637, p<0.001), disability subscale (correlation range r = -0.419 to -0.708, p<0.001), and total scores (correlation range r = -0404 to -0.697, p<0.001) showed statistically significant negative correlation to shoulder AROM, of which majority showed moderate correlation strength confirming our hypothesis. Forward flexion had the highest overall correlation with M-SPADI subscales and the total score (correlation range r = -0.637 to -0.708, p<0.001), while extension had the lowest correlation with M-SPADI subscales and the total score (correlation range r = -0.316 to -0.419, p<0.001) (Table 4).

*2. Known group validity.* The M-SPADI total scores were higher in participants with shoulder pain (Mdn: 33.8, IQR = 37.3) compared to no shoulder pain (Mdn: 0, IQR = 0.80), (U = 238.5, z = -13.9, p<0.001, r = 0.862).

**Table 4. Spearman's correlation coefficient between M-SPADI, NRS, and shoulder active range of motion.**

| | Spearman's correlation (r) | | |
|---|---|---|---|
| | **M-SPADI Pain** | **M-SPADI** **Disability** | **M-SPADI** **Total Score** |
| NRS | 0.845[1] | 0.722[1] | 0.795[1] |
| Forward Flexion | -0.637[1] | -0.708[1] | -0.697[1] |
| Abduction | -0.591[1] | -0.706[1] | -0.679[1] |
| Extension | -0.316[1] | -0.419[1] | -0.404[1] |
| Internal Rotation | -0.498[1] | -0.656[1] | -0.618[1] |
| External Rotation | -0.457[1] | -0.574[1] | -0.550[1] |

Internal and external rotation was measured at 90° abduction,

[1]p<0.001

M-SPADI: Malay Shoulder Pain and Disability Index, NRS: Numerical Rating Scale

Spearman's correlation strength: r = 1(perfect), r = 0.7–0.99 (strong), r = 0.4–0.69 (moderate) r = 0.1–0.39 (weak) and r = 0–0.09 (no correlation)

Overall Mann-Whitney U test revealed that M-SPADI pain scores, disability score, and total score were significantly higher in the shoulder pain group compared to the no shoulder pain group and the difference was statistically significant with a large effect size (Table 5).

**Floor and ceiling effect.** Less than 15% of participants with shoulder pain achieved the lowest or highest possible scores in the pain subscale (floor = 0%, ceiling = 0%), disability subscale (floor = 1.5%, ceiling = 0.8%), or total score (floor = 0%, ceiling = 0%) (Table 6).

**Table 5. Mann-Whitney U results comparing participants with and without shoulder pain.**

| M-SPADI | Group | Mean | SD | min-max | Median | IQR | Mann-Whitney U | Z score | p-value | Effect size (r) |
|---|---|---|---|---|---|---|---|---|---|---|
| Pain | Shoulder pain (n = 130) | 44.22 | 24.01 | 4–98 | 46.0 | 40.5 | 224.5 | -14.07 | <0.001 | 0.871 |
| | No shoulder pain (n = 130) | 1.75 | 5.78 | 0–48 | 0 | 0 | | | | |
| Disability | Shoulder pain (n = 130) | 32.05 | 23.24 | 0–100 | 26.3 | 37.8 | 499.5 | -13.60 | <0.001 | 0.843 |
| | No shoulder pain (n = 130) | 1.21 | 3.57 | 0–22 | 0 | 0 | | | | |
| Total Score | Shoulder pain (n = 130) | 36.71 | 22.46 | 2–87 | 33.8 | 37.3 | 238.5 | -13.9 | <0.001 | 0.862 |
| | No shoulder pain (n = 130) | 1.41 | 4.27 | 0–32 | 0 | 0.8 | | | | |

M-SPADI: Malay Shoulder Pain and Disability Index, SD: Standard Deviation, IQR: Interquartile range

Effect size: r≥0.20 (small), r≥0.50 (medium), r≥0.80 (large) [43, 44]

**Table 6. Floor and ceiling values, internal consistency, and test-retest reliability of M-SPADI.**

| M-SPADI | Min (Floor value%) | Max (Ceiling value%) | Cronbach's α | Intraclass Correlation Coefficient | | |
|---|---|---|---|---|---|---|
| | | | | ICC | 95% CI | P-value |
| Pain | 4 (0%) | 98(0%) | 0.914 | 0.922 | 0.867–0.954 | <0.001 |
| Disability | 0 (1.5%) | 100 (0.8%) | 0.945 | 0.859 | 0.759–0.917 | <0.001 |
| Total Score | 1.5 (0%) | 86.9 (0%) | - | 0.895 | 0.821–0.938 | <0.001 |

ICC: Intraclass correlation coefficient, CI: confidence interval

Floor value % and ceiling value %: frequencies of floor and ceiling values in percentage

Cronbach α = 0.70–0.95 is good internal consistency, Cronbach α ≥0.95 redundancy in items

ICC<0.500 poor, ICC = 0.500–0.749 moderate, ICC = 0.750–0.900 good, ICC >0.900 excellent.

### Reliability

**Internal consistency.** The M-SPADI pain and disability subscales had a high internal consistency with Cronbach's α = 0.914 and 0.945, respectively (Table 6).

**Test-retest reliability.** Fifty-six participants completed the test-retest with a mean time of 9.2±3.8 days between the first and second tests. The ICC results of the M-SPADI pain subscale, disability subscale, and total scores ($ICC_{Pain} = 0.922$, $ICC_{Disability} = 0.859$, $ICC_{Total} = 0.895$, $p < 0.001$) revealed good to excellent test-retest reliability [42].

## Discussion

The cross-cultural adaptation of M-SPADI adhered strictly to recommended guidelines [28, 29]. During the expert committee review, there were two main issues. The first was Disability Item 3: "putting on an undershirt or jumper," which scored I-CVI = 0.75. This item had issues with the word undershirt and jumper, thus requiring multiple amendments and a trial in the pretesting of M-SPADI before being accepted by the expert committee. This was a similar issue faced by the Brazilian-Portuguese study, which substituted the word "jumper" for "T-shirt" followed by the "term over your head" [13]. The pretesting of M-SPADI results showed no participant had issue with the item *memakai singlet atau baju sejuk*, and the subsequent decision by the expert committee and researchers was to accept the item. The second issue was ensuring M-SPADI was accessible and generalizable to all Malay-speaking individuals from various education and geographic backgrounds. This was accomplished by following the recommendation that PROMs should be simple enough for a 12- year old to understand [27]. Some initial Malay terms were accurate translations but were complicated and not commonly used. Simpler Malay words were selected; for example, *tengkuk* was replaced with *belakang leher*, and *mengenakan* was changed to *memakai*. Based on the pretesting of M-SPADI feedback and the Content Validity Index, M-SPADI has good face validity and content validity.

EFA showed that M-SPADI has a bi-dimensional structure with five items loading onto factor 1 and another five items loading onto factor 2. The three items which showed cross-loading were Pain Item 4, Disability Item 3, and Disability Item 6, all of which were overhead activities. Even though these items showed cross-loadings, they all had higher loadings on their predicted factors. The M-SPADI EFA revealed that items involving overhead activities tended to load for both pain and disability, while items regarding carrying heavy objects tend to load with pain despite being categorized as a disability item. Our EFA findings were similar to MacDermid et al. and may suggest that respondents cannot distinguish between pain and disability in some functional items as these two factors are closely related [45, 46].

Other studies which had similar EFA results of two factors with some items cross-loading were the original SPADI [1], Spanish SPADI [3], and Slovene SPADI [22]. The Chinese SPADI reported two distinct factors with no cross-loading, while other studies reported a single factor [23, 46] and up to 3 factors [19].

Convergent validity was established by comparing M-SPADI to the NRS and comparing M-SPADI to shoulder AROM. M-SPADI had a stronger positive correlation with the NRS when compared to the Spanish and Chinese SPADI studies [3, 4]. These studies both compared their respective SPADI scores to the VAS pain score, which resulted in moderate correlation for the pain subscale ($r_{Chinese/Spanish} = 0.488/0.670$), weak to moderate correlation strength for the disability subscale($r_{Chinese/Spanish} = 0.313/0.650$), and moderate correlation for total scores($r_{Chinese} = 0.402$) [3, 4]. Differing correlation strengths could be due to the studies utilising different pain scales which were the NRS and the VAS [3, 4].

Spearman's correlation proved the hypothesis that M-SPADI subscales and total scores were negatively correlated with shoulder AROM. Previous studies which performed similar

analyses include the Original SPADI [1], Telugu SPADI [16], and Tamil SPADI [17]. These studies reported a negative correlation between SPADI scores and shoulder AROM [1, 16, 17].

While all four abovementioned studies reported a negative correlation between SPADI scores and shoulder AROM, they reported different findings when comparing which scale had the highest and lowest correlation with shoulder AROM. M-SPADI disability subscale had the best correlation with AROM, whereas Telugu and Tamil SPADI pain subscale had the best correlation with AROM ($r_{Malay}$ = -0.416 to -0.708, $r_{Telugu}$ = -0.403 to -0.536, $r_{Tamil}$ = -0.455 to -0.585) [16, 17]. Regarding the lowest correlation, M-SPADI pain subscale, Telugu SPADI total score, and Tamil SPADI disability subscale scored the lowest correlation with AROM ($r_{Malay}$ = -0.316 to -0.637, $r_{Telugu}$ = -0.350 to -0.505, $r_{Tamil}$ = -0.482 to -0.588) [16, 17]. Original SPADI reported similar results for all three scales with moderate to strong correlation strength ($r_{original}$ = -0.520 to -0.803) [1]. Differences in the findings between M-SPADI, Tamil SPADI, and Telugu SPADI studies could be due to the differences in the severity and duration of shoulder disease at the time of data collection as the mean of the total score for M-SPADI was much lower compared to Telugu and Tamil (Mean$_{Malay}$:36.71, Mean$_{Telugu}$:81.31, Mean$_{Tamil}$:52.60) [16, 17]. Mean scores of AROM were also much higher in the M-SPADI study than the Telugu SPADI and Tamil SPADI study (Example: Abduction$_{Malay}$ = 134.61˚, Abduction$_{Telugu}$ = 83.00˚, Abduction$_{Tamil}$ = 103˚) [16, 17].

Regarding shoulder AROM, Original SPADI and Telugu SPADI had similar results where extension had the highest correlation with SPADI scales ($r_{Original}$ = -0.769 to -0.803, $r_{Telugu}$ = -0.386 to-0.536) [1, 16]. This differed from M-SPADI and Tamil SPADI, which reported forward flexion as having the highest correlation with SPADI scores ($r_{Malay}$ = -0.637 to -0.708, $r_{Tamil}$ = -0.577 to -0.588). A possible factor for these differing findings is the effect of different diagnoses on shoulder AROM. Most M-SPADI participants had rotator cuff injuries with or without impingement which we predict to have the most decrease in forward flexion and abduction and the least effect on extension. Comparatively, with diseases such as frozen shoulder, we expect a global reduction of range of motion.

Known-group validity testing demonstrated that the M-SPADI subscale and total scores were higher in the group with shoulder pain compared to the group without shoulder pain, and the difference was statistically significant. This confirmed that M-SPADI could discriminate between different groups, in this case, participants with and without shoulder pain. However, the authors acknowledge that the significant difference between the participants' age in the shoulder pain group and no shoulder pain group could be a confounding factor in this analysis. Other studies with reported known-group validity for SPADI include Dutch SPADI, Slovene SPADI, and Danish SPADI. They compared high and low initial pain scores [23], work absence versus no work absence [23], different severities of self-reported perceived disability [22], and working versus non-working participants [20].

Cronbach α for total score was not calculated as our EFA concluded that M-SPADI is bidimensional [26, 28]. Instead, we reported Cronbach α for each unidimensional domain: the pain subscale (Cronbach's α = 0.914) and the disability subscale (Cronbach's α = 0.945). These results are comparable to those reported in the original SPADI and other SPADI translations [1, 4, 13, 14, 16, 18]. These results show that the M-SPADI has a high internal consistency and does not have item redundancy as it did not cross the Cronbach's α>0.95 threshold.

The test-retest reliability for M-SPADI was rated good to excellent (ICC$_{Pain}$ = 0.922, ICC$_{Disability}$ = 0.859, ICC$_{Total}$ = 0.895) and was superior to the original SPADI, which had moderate test-retest reliability (ICC$_{Pain}$ = 0.655, ICC$_{Disability}$ = 0.644, ICC$_{Total}$ = 0.655). The Original SPADI findings could be due to its small sample size of 37 males. A recent systematic review reports test-retest reliability of SPADI ranges from ICC = 0.850–0.922, reflecting the findings of this study [9].

The strengths of this study were its large sample size fulfilling the COSMIN guidelines requirements [26, 28]. Moreover, this study adhered to the standard methods and guidelines for all procedures. In addition, this study includes known-group validity comparing participants with and without shoulder pain which has not been performed before.

The limitation of this study was that it was conducted in a single urban centre, with 75% of participants having tertiary education, which may cause bias in the results. A multicentre study conducted in rural and urban settings with participants of different educational backgrounds could yield different results. We also noted that the mean time interval for the test-retest was 9.2 ± 3.8 days, which was longer than the recommended seven days [9]. Like the Danish SPADI study, the M-SPADI results did not seem to be affected by this prolonged interval, and the results were in keeping with other SPADI translation studies [13, 14, 16–18, 20–22].

We recommend future prospective studies for the M-SPADI to examine measurement invariance, measurement error, and responsiveness in multicentre studies involving rural and urban populations.

## Conclusion

The M-SPADI has a bi-dimensional structure with good face and content validity, established construct validity, good internal consistency, and good to excellent test-retest reliability. It also has no floor or ceiling effect. Overall, the M-SPADI is a reliable and valid tool to assess pain and disability in Malay-speaking individuals with shoulder pain in clinical and research settings.

## Supporting information

**S1 Appendix. RedCap online demographics form.**
(DOCX)

**S2 Appendix. M-SPADI questionnaire.**
(DOCX)

**S3 Appendix. Content validity form.**
(DOCX)

**S4 Appendix. SPADI questionnaire.**
(DOCX)

## Acknowledgments

The authors would like to thank Associate Professor Dr. Mohamad Shariff Bin A Hamid (University of Malaya) for his guidance throughout this study.

## Author Contributions

**Conceptualization:** Caryn-Ann Ho, Jerri Chiu Yun Ling, Samihah Abdul Karim.

**Data curation:** Caryn-Ann Ho, Jerri Chiu Yun Ling, Samihah Abdul Karim.

**Formal analysis:** Caryn-Ann Ho, Jerri Chiu Yun Ling, Samihah Abdul Karim.

**Funding acquisition:** Samihah Abdul Karim.

**Investigation:** Caryn-Ann Ho, Jerri Chiu Yun Ling, Samihah Abdul Karim.

**Methodology:** Caryn-Ann Ho, Jerri Chiu Yun Ling, Samihah Abdul Karim.

**Project administration:** Jerri Chiu Yun Ling.

**Resources:** Caryn-Ann Ho, Jerri Chiu Yun Ling, Samihah Abdul Karim.

**Supervision:** Jerri Chiu Yun Ling.

**Validation:** Caryn-Ann Ho, Jerri Chiu Yun Ling, Samihah Abdul Karim.

**Visualization:** Jerri Chiu Yun Ling, Samihah Abdul Karim.

**Writing – original draft:** Caryn-Ann Ho, Jerri Chiu Yun Ling, Samihah Abdul Karim.

**Writing – review & editing:** Caryn-Ann Ho, Jerri Chiu Yun Ling, Samihah Abdul Karim.

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
