## [Decision Letter · Decision Letter 0]

5 Oct 2021

PONE-D-21-24314Cross-cultural adaptation and measurement properties of the Malay Shoulder Pain and Disability IndexPLOS ONE

Dear Dr. Abdul Karim,

Thank you for submitting your manuscript to PLOS ONE. After careful consideration, we feel that it has merit but does not fully meet PLOS ONE’s publication criteria as it currently stands. Therefore, we invite you to submit a revised version of the manuscript that addresses the points raised during the review process.

 Reviewer two also provided comments in the attached document.

We look forward to receiving your revised manuscript.

Kind regards,

Stefan Hoefer

Academic Editor

PLOS ONE

Journal Requirements:

2. PLOS ONE has specific requirements for studies that are presenting a new method or tool as the primary focus, including a newly developed or modified questionnaire or scale (https://journals.plos.org/plosone/s/submission-guidelines#loc-methods-software-databases-and-tools). One requirement is that the questionnaire or scale must be openly available under a license no more restrictive than CC BY. In light of this, before we proceed, we request that you comply with the following requests: 

1)Please provide details as to which scale items, if any, are derived from published scales, and provide the DOI and full reference for any articles from which you used or adapted/translated scale items.

2)If your scale includes items that are copied directly or adapted/translated from previously published tools those items may be subject to copyright restrictions. If the original scale was not published in an open access article under a CC BY license we ask that you contact the original copyright holder of the scale you modified/translated to obtain permission to publish the scale described in the manuscript in PLOS ONE under CC-BY. Please note that the copyright may not necessarily be held by the authors of the previous study, and may belong to the publisher.

We recommend that you contact the original copyright holder with the PLOS Content Permission Form (http://journals.plos.org/plosone/s/file?id=7c09/content-permission-form.pdf) and the following text:

“I request permission for the open-access journal PLOS ONE to publish XXX under the Creative Commons Attribution License (CCAL) CC BY 4.0 (http://creativecommons.org/licenses/by/4.0/). Please be aware that this license allows unrestricted use and distribution, even commercially, by third parties. When you reply, please state clearly whether you grant permissions to republish this content under the CC BY 4.0 license, and provide a completed version of the attached form. Please note that if you cannot grant permission to republish XX in an open access article under the CC BY license I may cancel this permissions request. ”

Please upload the completed Content Permission Form(s) or other proof of granted permissions as an "Other" file with your submission. Please note that if the permissions documentation you obtain from the original article’s copyright holder does not explicitly specify permissions to publish the material under the CC BY license, we will not be able to proceed with your submission. **Please note that ‘Rightslink’ documents are not typically compatible with publication of your scale.**

3)If you obtain permissions to publish the items under the CC BY license, please add the following to the caption of the questionnaire items: “Modified from [ref] under a CC BY license, printed with permission from [name of copyright holder], original copyright [original copyright year].”

Thank you for your attention to this query. If you have questions, please contact us at plosone@plos.org. We look forward to receiving your resubmitted manuscript.

Reviewers' comments:

Reviewer's Responses to Questions

**Comments to the Author**

1. Is the manuscript technically sound, and do the data support the conclusions?

Reviewer #1: Yes

Reviewer #2: Yes

2. Has the statistical analysis been performed appropriately and rigorously? 

Reviewer #1: Yes

Reviewer #2: Yes

3. Have the authors made all data underlying the findings in their manuscript fully available?

Reviewer #1: Yes

Reviewer #2: Yes

4. Is the manuscript presented in an intelligible fashion and written in standard English?

Reviewer #1: Yes

Reviewer #2: Yes

5. Review Comments to the Author

Reviewer #1: The manuscript was written clearly. The reported study was done according the proper procedures and standards.

1. "to cross-culturally adapt the Malay version of the Shoulder Pain 25 and Disability Index (M-SPADI)" This sentence suggest that a Malay version of SPADI is already exist. I suggest this sentence to be revised to more accurately describe the objective of the study.

2. For the respondents' inclusion criteria, was the presence of other types of pains or illnesses considered? Can the authors argue that the presence of other illnesses would not affect the findings?

3. From where was the no-pain group sampled? Are their demographic characteristics comparable to the pain group?

4. What were measured in the pilot study? The results section mentioned about 'item relevance', but this was not mentioned clearly in the Method section.

5. 'Pilot of prefinal' sounds a bit awkward. 'Pretesting of M-SPADI' may be more accurate. However, I do acknowledge that this comment is more about semantic preferences, and not about study design technicality.

6. Why was S-W selected for test of normality? Given the sample size, is K-S not a better option?

7. Tests of normality were mentioned in Methods section, but not reported in Results section. Or perhaps I missed them.

8. Describe the experts for the content validation study.

Reviewer #2: 1. No report on psychometric properties or validity findings from previous studies in the materials/ measure section.

2. Should report comparability & interpretability score to show agreement between examiners on the tool translation.

6. PLOS authors have the option to publish the peer review history of their article (what does this mean?). If published, this will include your full peer review and any attached files.

Reviewer #1: No

Reviewer #2: No

---

## [Author Response · Author response to Decision Letter 0]

6 Nov 2021

Thank you for the valuable comments of this manuscript. 

Reviewer #1:

1. "to cross-culturally adapt the Malay version of the Shoulder Pain 25 and Disability Index (M-SPADI)" This sentence suggest that a Malay version of SPADI is already exist. I suggest this sentence to be revised to more accurately describe the objective of the study.

Response: Thank you for this suggestion and we agree with your assessment. Accordingly, we have amended the sentence as follows.

Changes in the manuscript: 

The purpose of this study is to cross-culturally adapt the Shoulder Pain and Disability Index from English to Malay and to evaluate the measurement properties of the Malay version among Malay speakers with shoulder pain. (Line 24-26, Abstract, Objective)

Our objective was to cross-culturally adapt the Shoulder Pain and Disability Index from English to Malay, and to evaluate the measurement properties of the Malay version among Malay speakers with shoulder pain. (Line 89-91, Introduction)

2. For the respondents' inclusion criteria, was the presence of other types of pains or illnesses considered? Can the authors argue that the presence of other illnesses would not affect the findings?

Response: Thank you for this comment and we agree that other conditions and injuries would affect this study’s findings. To avoid this, the exclusion criteria for both the shoulder pain group and no shoulder pain group were neck, elbow, wrist, or hand injury, decline to give consent and psychiatric illness. We regret that this was not clear in the original manuscript and have amended the paragraph under the Participants heading and further clarified the exclusion criteria as follows.

Changes in the manuscript: 

The inclusion criteria for the shoulder pain group were patients attending Sports Medicine Clinic with shoulder pain, ≥ 18 years old, and understood the Malay language. The inclusion criteria for the no shoulder pain group were patients attending Sports Medicine Clinic without shoulder pain, ≥ 18 years old, and understood the Malay language. For both shoulder pain and no shoulder pain group, the exclusion criteria were neck, elbow, wrist, or hand injury, decline to give consent, and psychiatric illness. (Line 110-115, Materials and methods, Participants)

3. From where was the no-pain group sampled? Are their demographic characteristics comparable to the pain group?

I. From where was the no-pain group sampled?

Response: Thank you for the question. The no shoulder pain group was sampled from patients attending the Sports Medicine Clinic, UMMC from 1st June 2020 to 12th May 2021. We realize that this was not clearly stated under participants and have made the following amendments.

Changes in the manuscript: 

Research participants for the shoulder pain group and the no shoulder pain group were recruited via universal sampling from patients attending the Sports Medicine Clinic, UMMC from 1st June 2020 to 12th May 2021. (Line 108-110, Materials and methods, Participants)

II. Are their demographic characteristics comparable to the pain group?

Response: Thank you for raising this important point. Based on this comment and comment no 7, we recalculated the normality testing with Kolmogorov-Smirnov test then ran Mann-Whitney U test to assess if the demographic characteristics of the shoulder pain group and no shoulder pain group were significantly different. The results are as follows.

Changes in the manuscript: We redrafted the Description of sample paragraph and added

The demographic characteristics of the shoulder pain group and no shoulder pain group differed significantly in age (shoulder pain group: median=55.8, Interquartile range, IQR=24.6, no shoulder pain group: median=35.27, IQR=23.3, P<0.05). However, there was no statistically significant difference between gender and level of education between the shoulder pain and no shoulder pain group (p>0.05). (Line 245-249, Results, Description of sample)

We also further clarified in the discussion that 

the authors acknowledge that the significant difference between the participants’ age in the shoulder pain group and no shoulder pain group could be a confounding factor in this analysis. (Line 476-477, Discussion)

4. What were measured in the pilot study? The results section mentioned about 'item relevance', but this was not mentioned clearly in the Method section.

Response: Thank you for highlighting this issue. Following comment 6, we have standardized our terminology and have replaced the terms “pilot of prefinal” and “pilot study” with “pretesting of M-SPADI.”

In pretesting of M-SPADI we asked for feedback from 50 participants regarding the following questions:

1. What did they understand about the questions asked (Both item and response)?

2. Were the questions relevant to shoulder pain and disability? (Yes/No)

3. Was there any difficulty understanding the questions? (Yes/No)

4. Any suggestions for improvement?

Pretesting of M-SPADI is stage 5 in the Beaton et al guidelines for Cross-cultural Adaptation of Self-Report Measure (27). It provides insight on how participants interpret the items and roughly ensures that the adapted version retains its equivalence (27,30). We subsequently used the results of the questions in pretesting of M-SPADI to assess Face Validity.

We regret that this was not written clearly and have made the following amendments. First, to the Stage 5: Pretesting of M-SPADI methodology and second, to the Face Validity results.

Changes in the Manuscript:

Fifty participants completed the prefinal version of M-SPADI and answered the following questions; What did they understand about the questions asked (Both items and response)?(27) Were the questions relevant to shoulder pain and disability?(30) Was there any difficulty understanding the questions? and Any suggestions for improvement? These answers were subsequently used to assess Face Validity. (Line 177-181, Materials and methods, Stage 5: Pretesting of M-SPADI)

The feedback from the pretesting of M-SPADI was good, with minimal issues in understanding the M-SPADI items and response by participants. All participants agreed that the questionnaire was clear and easy to understand (100%), and 86% gave positive feedback when asked if the questions were relevant to shoulder pain and disability. Overall M-SPADI has good face validity. (Line 298-302, Results, Face Validity)

5. 'Pilot of prefinal' sounds a bit awkward. 'Pretesting of M-SPADI' may be more accurate. However, I do acknowledge that this comment is more about semantic preferences, and not about study design technicality.

Response: Thank you for your suggestion. We agree with you and have incorporated this suggestion throughout our paper by standardizing our terminology. The terms “pilot of prefinal” and “pilot study” have been replaced with “pretesting of M-SPADI”. 

Changes in the manuscript: The terms “pilot of prefinal” and “pilot study” have been replaced with “pretesting of M-SPADI.”

6. Why was S-W selected for test of normality? Given the sample size, is K-S not a better option?

Response: Thank you for this comment. We agree that Kolmogorov-Smirnov is a better option based on our large sample size and have re-run our normality calculations with this test. We have amended our manuscript to reflect this change. The results of the Kolmogorov-Smirnov testing were similar to the Shapiro-wilk and no further changes in the statistical analysis was necessary.

Changes in the manuscript: 

normality for all data was assessed using Q-Q plots and the Kolmogorov-Smirnov test. (Line 206-207, Materials and methods, Statistical Analysis)

7. Tests of normality were mentioned in Methods section, but not reported in Results section. Or perhaps I missed them.

Response: Thank you for this comment, we regret the oversight of not reporting our results of normality testing. On further review we also noted that we reported non normally distributed data (participant age, M-SPADI scores, NRS scores and active range of motion) in mean and standard deviation. We have added the results of normality testing in the Description of Sample, Results section and amended table 1 as detailed below. 

Changes in the manuscript: 

Q-Q Plots and Kolmogorov-Smirnov test (p<0.05) revealed that data for M-SPADI scores, NRS scores and shoulder AROM were non normally distributed for the pretesting of M-SPADI group, the shoulder pain group, and the no shoulder pain group. (Line 250-253, Results, Description of sample)

For Table 1, the participants’ age, M-SPADI scores, NRS scores, shoulder AROM scores were changed from mean and standard deviation to median and interquartile range. (Line 265-266, Results, Description of sample)

8. Describe the experts for the content validation study.

Response: Thank you for this comment. The experts for the content validation study are described in Stage 4: Expert committee. We have made a few amendments as detailed below to clarify this point.

Changes in the manuscript: 

The expert committee consisted of a senior lecturer cum certified translator from the University Malaya Faculty of Languages and Linguistics, a Professor of Orthopaedic Surgery, a Professor of Family Medicine, two Sports Medicine Physicians, a Rehabilitation Medicine Physician, a Senior Physiotherapist, an Occupational Therapist, and an Exercise Physiologist. They also completed the content validity forms for content validity assessment. (Line 169-174, Materials and methods, Stage 4: Expert committee)

Content validity was evaluated by a panel of eight expert committee members using Content validity forms (Appendix 4) during stage 4: Expert committee. (Line 211-212, Statistical Analysis, Validity)

Reviewer #2:

Thank you, reviewer 2 for your review and comments of this manuscript.

1. No report on psychometric properties or validity findings from previous studies in the materials/ measure section.

Response: Thank you for this comment. We agree with you and have included a report on psychometric properties of SPADI under the Shoulder Pain and Disability Index, Instruments heading as suggested.

Changes in the manuscript: 

The original SPADI development study reported good internal consistency with Cronbach α total=0.95, pain=0.86 and disability=0.95(1). They also reported a moderate test-retest reliability with intraclass correlation coefficient (ICC)=0.66 and moderate to high negative correlation between shoulder active range of motion (AROM) and SPADI scores (Pearson correlation coefficient range, r= -0.545 to -0.804, p<0.001) (1). A recent systematic review reported that SPADI had a bidimensional structure, good internal consistency with Cronbach α ranging from 0.85-0.96, good to excellent test-retest reliability with ICC ranging from 0.89- 0.92 and no floor or ceiling effects (9). (Line 127-135, Instruments, Shoulder Pain and Disability Index)

2. Should report comparability & interpretability score to show agreement between examiners on the tool translation.

Response: Thank you for your comment. We acknowledge that assessing comparability and interpretability score during the cross-cultural adaptation process would increase the strength and quality of this manuscript. However, our study design followed the 6 step, guidelines for the process of cross-cultural adaptation by Beaton et al 2000 which has also been adapted by the American Association of Orthopedic Surgeons Outcomes Committee (27). We regretfully are unable to report this result as it was not in the original study design as recommended by Beaton et al 2000.

Changes in the manuscript: None

3. Mistake highlighted in line 39 p-value<0.001

Response: Thank you for highlighting this issue.

Changes in the manuscript: We have changes p-value<0.001 to p<0.001. (Line 39, Abstract, Results)

4. Disability subscale item 3 I-CVI 0.75: Should you consider deleting & replacing this item?

Response: Thank you for raising this important question. Based on our reference Davis et al, since Disability Item 3’s I-CVI score was 0.75 which is in between 0.70-0.79 it required amendment but not deletion (38). The item was amended, and then tested during the pretesting of M-SPADI. Based on the pretesting of M-SPADI results which showed no participant had any issue with the amended item the expert committee agreed to keep the amended Disability Item 3.

The authors acknowledge that an I-CVI score of less than 0.79 is not ideal and have highlighted this issue and its resolution in the Discussion section as highlighted below. 

The first was Disability Item 3: "putting on an undershirt or jumper," which scored I-CVI=0.75. This item had issues with the word undershirt and jumper, thus requiring multiple amendments and a trial in the pretesting of M-SPADI before being accepted by the expert committee. This was a similar issue faced by the Brazilian-Portuguese study, which substituted the word "jumper" for "T-shirt" followed by the "term over your head" (13). The pretesting of M-SPADI results showed no participant had issue with the item ‘memakai singlet atau baju sejuk’ and the subsequent decision by the expert committee and researchers was to accept the item. (Line 408-415, Discussion) 

We have also amended our Content Validity, Results section to add further clarification regarding this issue.

Changes in the manuscript:

Since Disability Item 3’s I-CVI score was between 0.70-0.79 it required amendment but not elimination(38). Following the content validity scoring, this item was amended by the expert committee and trialled during the pretesting of M-SPADI. Based on the pretesting of M-SPADI results the expert committee agreed to keep the item. (Line 283-287 Results, Content Validity)

---

## [Decision Letter · Decision Letter 1]

28 Feb 2022

Cross-cultural adaptation and measurement properties of the Malay Shoulder Pain and Disability Index

PONE-D-21-24314R1

Dear Dr. Abdul Karim,

We’re pleased to inform you that your manuscript has been judged scientifically suitable for publication and will be formally accepted for publication once it meets all outstanding technical requirements.

Kind regards,

Fatih Özden, PhD

Academic Editor

PLOS ONE

Reviewers' comments:

Reviewer's Responses to Questions

**Comments to the Author**

1. If the authors have adequately addressed your comments raised in a previous round of review and you feel that this manuscript is now acceptable for publication, you may indicate that here to bypass the “Comments to the Author” section, enter your conflict of interest statement in the “Confidential to Editor” section, and submit your "Accept" recommendation.

Reviewer #1: All comments have been addressed

2. Is the manuscript technically sound, and do the data support the conclusions?

Reviewer #1: Yes

3. Has the statistical analysis been performed appropriately and rigorously? 

Reviewer #1: Yes

4. Have the authors made all data underlying the findings in their manuscript fully available?

Reviewer #1: Yes

5. Is the manuscript presented in an intelligible fashion and written in standard English?

Reviewer #1: Yes

6. Review Comments to the Author

Reviewer #1: The authors have written the amendments satisfactorily. There are no further issue that needs to be addressed.

7. PLOS authors have the option to publish the peer review history of their article (what does this mean?). If published, this will include your full peer review and any attached files.

Reviewer #1: No

---

## [Editor Report · Acceptance letter]

4 Mar 2022

PONE-D-21-24314R1 

Cross-cultural adaptation and measurement properties of the Malay Shoulder Pain and Disability Index 

Dear Dr. Abdul Karim:

I'm pleased to inform you that your manuscript has been deemed suitable for publication in PLOS ONE. Congratulations! Your manuscript is now with our production department. 

Kind regards, 

on behalf of

Dr. Fatih Özden 

Academic Editor

PLOS ONE